# Role of Pre-Operative Brain Imaging in Patients with NSCLC Stage I: A Retrospective, Multicenter Analysis

**DOI:** 10.3390/cancers14102419

**Published:** 2022-05-13

**Authors:** Luis Filipe Azenha, Pietro Bertoglio, Peter Kestenholz, Michel Gonzalez, Matyas Pal, Thorsten Krueger, Bassam Redwan, Volkan Koesek, Eyad Al Masri, Takuro Miyazaki, Farahnaz Sadegh Beigee, Benedetta Bedetti, Philipp Schnorr, Joachim Schmidt, Patrick Zardo, Laura Boschetti, Sven Oliver Schumann, Fabrizio Minervini

**Affiliations:** 1Department of Thoracic Surgery, Cantonal Hospital Lucerne, 6000 Lucerne, Switzerland; filipe.azenha@luks.ch (L.F.A.); peter.kestenholz@luks.ch (P.K.); 2Division of Thoracic Surgery, IRCSS Azienda Ospedaliero-Universitaria, 40138 Bologna, Italy; pietro.bertoglio@aosp.bo.it; 3Department of Thoracic Surgery, Lausanne University Hospital, 1011 Lausanne, Switzerland; michel.gonzalez@chuv.ch (M.G.); matyas.pal@chuv.ch (M.P.); thorsten.krueger@chuv.ch (T.K.); 4Department of Thoracic Surgery, Westfalen Hospital, 44536 Lünen, Germany; bassam.redwan@klinikum-westfalen.de (B.R.); volkan.koesek@klinikum-westfalen.de (V.K.); eyad.almasri@klinikum-westfalen.de (E.A.M.); 5Department of Surgical Oncology, Nagasaki University, Nagasaki 852-8102, Japan; miyataku@nagasaki-u.ac.jp; 6Department of Thoracic Surgery, Shahid Beheshti University, Tehran 69411, Iran; beigeef@sbmu.ac.ir; 7Department of Thoracic Surgery, Helios Hospital, 53123 Bonn Rhein-Sieg, Germany; benedetta.bedetti@helios-gesundheit.de (B.B.); philipp.schnorr@helios-gesundheit.de (P.S.); joachim.schmidt@helios-gesundheit.de (J.S.); 8Division of Thoracic Surgery, Department of General, Thoracic and Vascular Surgery, Bonn University Hospital, 53127 Bonn, Germany; 9Department of Cardiothoracic Surgery, Medical School Hannover, 30625 Hannover, Germany; patrick.zardo@mh-hannover.de; 10Division of Oncology, Cantonal Hospital Lucerne, 6210 Sursee, Switzerland; laura.boschetti@luks.ch; 11Department of General Internal Medicine, Bülach Hospital, 8180 Bülach, Switzerland; svenoliverschumann@web.de

**Keywords:** lung cancer, brain imaging, MRI, NSCLC

## Abstract

**Simple Summary:**

Lung cancer is the worldwide leading cause of cancer-related death among both genders, with about 230,000 patients/year being diagnosed in the US alone. It accounts for about 40% of all brain metastases, which already occur in around 3% of early-stage patients. Nonetheless, current international guidelines do not unanimously recommend brain imaging for use in the early stages of cancer. Some studies have suggested that surgical or radiosurgical treatment of brain metastases may provide better survival, especially in asymptomatic patients. Additionally, advances in genome analysis have identified molecular targets for pharmaceutical agents. These recent advancements in treatment stress the importance of studying incidence as well as patient and tumor characteristics in order to potentially adapt future guidelines and provide the best possible treatment for early-stage lung cancer. This multicentric study analyzed the data of 577 patients diagnosed with early-stage lung cancer who had been submitted for brain imaging at initial tumor staging.

**Abstract:**

*Background:* Lung cancer is the worldwide leading oncological cause of death in both genders combined and accounts for around 40–50% of brain metastases in general. In early-stage lung cancer, the incidence of brain metastases is around 3%. Since the early detection of asymptomatic cerebral metastases is of prognostic value, the aim of this study was to analyze the incidence of brain metastases in early-stage lung cancer and identify possible risk factors. *Methods:* We conducted a retrospective multicentric analysis of patients with Stage I (based on T and N stage only) Non-Small Cell Lung Cancer (NSCLC) who had received preoperative cerebral imaging in the form of contrast-enhanced CT or MRI. Patients with a history of NSCLC, synchronous malignancy, or neurological symptoms were excluded from the study. Analyzed variables were gender, age, tumor histology, cerebral imaging findings, smoking history, and tumor size. Results were expressed as mean with standard deviation or median with range. *Results:* In total, 577 patients were included in our study. Eight (1.4%) patients were found to have brain metastases in preoperative brain imaging. Tumor histology was adenocarcinoma in all eight cases. Patients were treated with radiotherapy (five), surgical resection (two), or both (one) prior to thoracic surgical treatment. Other than tumor histology, no statistically significant characteristics were found to be predictive of brain metastases. *Conclusion:* Given the low incidence of brain metastases in patients with clinical Stage I NSCLC, brain imaging in this cohort could be avoided.

## 1. Introduction

Lung cancer is the worldwide leading oncological cause of death in both genders combined [1,2]. Around 44,500 people are diagnosed with lung cancer each year in the UK and more than 230,000 are diagnosed each year in the USA [3]. Additionally, Non-Small Cell Lung Cancer (NSCLC) accounts for about 40–50% of brain metastases, making it the most frequent cause of brain metastases [4,5]. Brain metastases will affect 40–50% of patients during the course of the disease and have a significant impact on their survival as well as their quality of life [6]. The most common treatment options for patients with NSCLC include surgery, radiotherapy, chemotherapy, and immunotherapy [7]. With the advances made in tumor genome analysis, several oncogenic driver mutations have been identified. Epidermal growth factor receptor (EGFR), ROS1, and ALK translocation have given us the possibility of providing targeted treatment [8]. Accurate staging is the cornerstone for appropriate treatment in patients with NSCLC. The importance of neuroimaging for locally advanced or symptomatic patients is clear, but in early-stage asymptomatic NSCLC patients, many physicians instead perform brain MRI or CT scans, although recommendations are lacking [9,10]. The overall survival of patients with brain metastases is generally poor and ranges from 6 to 9 months, although some studies suggest that the detection of asymptomatic brain metastases is a prognostic factor in terms of survival [4]. The aim of this study is to analyze the incidence of brain metastases in patients with Stage I NSCLC as well as tumor and patient characteristics.

## 2. Materials and Methods

This study was conducted according to the guidelines of the Declaration of Helsinki and approved by the Ethics Board of Central Switzerland (28 October 2021, project #2021-02034). This article was written according to the strengthening the reporting of observational studies in epidemiology (STROBE) guidelines [11].

### 2.1. Patients

This study was designed as a multicentric retrospective analysis. We collected data from all patients older than 18 years affected by a clinical Stage IA-IB-IC NSCLC diagnosed between January 2019 and December 2020 in seven thoracic surgery units (Lucerne, Lausanne, Hannover, Lünen, Bonn, Teheran and Nagasaki). We included patients with a NSCLC clinical Stage I (based only on T and N status) who received brain imaging in the preoperative workup, either by contrast-enhanced CT scan or MRI. We excluded from the analysis all patients with any prior history of lung cancer, further synchronous or metachronous disease, previous brain malignancies, or neurological symptoms requiring a priori pre-operative brain imaging.

### 2.2. Endpoints

We aimed to assess the incidence of brain metastasis in clinical early-stage, node-negative NSCLC and to evaluate the possible features of patients who developed brain metastases. 

### 2.3. Statistical Analysis

Data were analyzed using the software SPSS version 23.0 for Windows (SPSS Inc., Chicago, Ill., USA). Continuous variables were expressed in terms of mean with standard deviation (SD) or median with range, while categorical variables were expressed in terms of frequency. The two-tailed Pearson’s chi-square test and likelihood ratio test were used for the intergroup comparison of categorical variables, while the Student’s *t* test was used for continuous variables. A *p*-value < 0.05 was considered statistically significant.

## 3. Results

Table 1 reports the main features of our cohort. Briefly, we included in the study 577 patients; 320 (55.5%) were male and the mean age was 68.8 years (±SD 9.0). A total of 456 patients (79%) had a history of smoking. Adenocarcinoma was the most frequent diagnosis (400 patients, 69.3%), followed by squamous cell carcinoma (135, 23.4%), while the remaining 42 cases (7.3%) had different histologies. In total, 480 patients (83.2%) underwent MRI, while 97 (16.8%) underwent contrast-enhanced CT scan. The vast majority of our cohort (569, 98.6%) had no brain metastases. In eight cases with brain metastases, four had a single metastasis, two metastases were found in two cases, and three and for metastases were found in one case each.

Among these patients, the majority were female (5, 62.5%), with a mean age of 63.5 years (±SD 6.4). All of them had a smoking history, with a mean pack/year of 43.7 (±SD 9.3). Half of these patients had a previous malignancy in their medical history. Most patients (5, 62.5%) received radiotherapy for their brain metastases, while surgery alone was performed in two (25.0%) cases and the remaining received surgery and radiotherapy. All of them received lung resection after the treatment of brain metastasis. We compared different features of patients with and without brain metastases, despite the small number of patients with brain metastases identified, and did not find any statistically significant difference between both groups when we considered age (*p* = 0.096), gender (*p* = 0.303), smoking history (*p* = 0.185), presence of previous malignancies (*p* = 0.123), side of the lung cancer (*p* = 0.899), SUVmax of the lung cancer (*p* = 0.637), tumor dimension (*p* = 0. 532), and platelet–lymphocyte ratio (PLR, *p* = 0.954). On the other hand, all patients with brain metastasis were diagnosed with lung adenocarcinoma (*p* = 0.038).

## 4. Discussion

Our retrospective analysis showed that 1.4% of patients with clinical Stage I NSCLC presented with brain metastases, which is in line with previous studies suggesting that cerebral imaging in stage I NSCLC is widely overused [9]. Silent brain metastases have been reported to occur in roughly 30% of patients diagnosed with NSCLC (>T1N0M0). This number drops to around 3% in Stage I NSCLC, which is why brain imaging in this subgroup of patients is not recommended, as it may result in increased costs and delays in therapy and have only a marginal impact on patient management [12]. Balekian et al., using the data available from the National Lung Screening trial, found that one in eight patients with clinical Stage IA disease underwent brain MRI or CT, despite a lack of guidelines supporting this practice. Interestingly, none of the patients in this analysis had brain metastases [13]. A prospective study which included 91 patients found an incidence of brain metastases of 3% in Stage I-II and 20% in Stage IIIA NSCLC patients who underwent brain imaging screening [14]. Yohena and colleagues reported a low percentage of brain metastases in resectable NSCLC after MRI screening (0% in N0-disease, 5.2% in N1-disease, and 4.7% in N2-disease) [15]. Despite these reports, there is some evidence supporting the use of brain imaging in patients scheduled for lung resection in NSCLC. O’Dowd and colleagues reported that patients with lung malignancy who developed metastases in the further course of their disease are most likely to be those initially discovered at an early stage (73% stage I-II) [16]. Hudson et al. still recommend the use of brain imaging prior to lung resection, despite a low percentage of 5.3% having brain metastases. They state that, even if the incidence is low, newly diagnosed brain metastases potentially change the management of these patients [17].

Rami-Porta et al. recommend cerebral staging in all patients with curative therapeutic options apart from in patients with pure ground glass adenocarcinoma [18]. This recommendation is supported by Postmus et al., who established the ESMO Clinical Practice Guidelines for the diagnosis, treatment, and follow-up for early and locally advanced NSCLC [19]. On the other hand, NCCN and the British Thoracic Society and the National Institute for Health and Care Excellence guidelines recommend the use of brain MRI for Stage II-III and declare it optional for stage IB but not for stage IA NSCLC [20,21].

As contrast-enhanced MRI has a superior sensitivity in brain metastasis detection compared to contrast-enhanced CT [22,23,24] and 81.3% of patients enrolled in this study underwent MRI, we consider our primary endpoint well addressed in our cohort. Additionally, contrast-enhanced CT scan has a negative predictive value of 94% when it comes to the diagnosis of brain metastasis [25]. Tumor size, histology, and N stage have been reported to be risk factors for brain metastases [26]. In our study population, only patients with adenocarcinoma were diagnosed with brain metastases, in accordance with previous studies that confirm adenocarcinoma as a risk factor [27,28,29]. In the end, brain imaging appears to be redundant in patients with Stage I NSCLC, which is in line with the Choosing Wisely campaign partnered with the Society of Thoracic Surgeons (STS) in the United States.

The development of genome analysis and the detection of several oncogenic driver mutations such as mutations in EGFR, ALK, and ROS 1 have led to the concept of personalized medicine. These biomarkers represent targets for therapeutic agents with the consequent improvement of overall survival in patients with NSCLC and brain metastasis [8]. Hence, it is ethically questionable whether patients with early-stage NSCLC should be deprived of brain imaging even though a small proportion present brain metastasis at an early stage and subsequently would not receive the best possible treatment. Since it has been established that patients with EGFR mutation are more likely to present brain metastases, molecular analysis of tissue samples pre- or postoperatively could be used as a tool to determine which patients are most likely to benefit from brain imaging [30]. Of course, this possibility is reserved for cases in which histology has been determined preoperatively.

The limitations of our study mainly relate to its retrospective design, which is based on reviews of data primarily collected for clinical or quality control purposes. The multicenter design leads simultaneously to a large variability of patients but to a broader sample, thus reflecting a real-life scenario with potentially more generalizable conclusions.

## 5. Conclusions

The utility of pre-operative brain imaging in patients with Stage I NSCLC undergoing lung resection is a controversial topic. This is reflected by the different guidelines established by leading oncological societies worldwide, such as NCCN, ESMO, NICE, and BTS. Our data suggest that the incidence of brain metastasis in clinical early-stage lung cancer is low and that brain imaging might be avoided in this subset of patients. On the other hand, if the histology is known before surgery, the advances in the targeted treatment of NSCLC with the identification of molecular drivers could help us to define a subset of patient with clinical stage I who could benefit from pre-operative brain imaging.

## Figures and Tables

**Table 1 cancers-14-02419-t001:** Patient demographics.

Gender *n* (%) Male	320 (55.5)
**Age mean (±SD)**	68.8 (±9.0)
**BMI mean (±SD)**	24.7 (±4.6)
**Smoking history *n* (%)** Yes (current or previous smoker)	456 (79.0)
**Pack/year mean (±SD)** (only for patients with a smoking history)	35.8 (±28.4)
**Previous malignancies n (%)** Yes	151 (26.2)
**Clinical tumor dimension mean (±SD)** mm	20.8 (± 9.0)
**SUV mean (±SD)**	6.6 (±5.1)
**PLR mean (±SD)**	100.2 (78.1)
**Histology *n* (%)** Adenocarcinoma Squamous Cell carcinoma Other histologies	400 (69.3) 135 (23.4) 42 (7.3)

BMI: Body mass index; SD: standard deviation; SUV: standardized uptake value; PLR: platelet to lymphocyte ratio.

## Data Availability

The data presented in this study are available on request from the corresponding author. The data are not publicly available due to private and ethical restrictions.

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
