# Peer review of "Role of Pre-Operative Brain Imaging in Patients with NSCLC Stage I: A Retrospective, Multicenter Analysis"

_cancers, 2022, doi:10.3390/cancers14102419_

Round 1

Reviewer 1 Report

  • Row 41 correct “NSCL”
  • Row 59: include reference “Nayak L, Lee EQ, Wen PY: Epidemiology of brain metastases. Curr Oncol Rep 14:48-54, 2012” for epidemiology of brain mets in NSCLC, not reference number 4
  • Row 61-62: therapies for metastatic NSCLC include also targeted agents.
  • Specify different incidence of brain metastases in oncogene addicted NSCLC (higher incidence in patients with EGFR, ALK, ROS1 driver).
  • ESMO guidelines suggest screening brain MRI with level of recommendation IIIB (Postmus PE et al. Annals of Oncology 28 (Supplement 4): iv1–iv21, 2017 doi:10.1093/annonc/mdx222).
  • NNCN guidelines recommend brain MRI for stage II-III, while it is optional for stage IB, and not recommended for stage IA (NCCN version 3.2022)
  • Were observed any differences among stage IA, IB and IC?
  • Considering different incidence of brain metastases in patients with oncogene addicted NSCLC, data about oncogene drivers and PD-L1 expression are needed.
  • Reference number 6 must be updated: ESMO guidelines Annals of Oncology 29 (Supplement 4): iv192–iv237, 2018 doi:10.1093/annonc/mdy275

Author Response

Dear reviewer,

thank you for your constructive comments concerning our paper. We corrected the passages you pointed out and added the literature you suggested.

Best regards,

The authors

  • Row 41 correct “NSCL”
    • Line 45 NSCLC
  • Row 59: include reference “Nayak L, Lee EQ, Wen PY: Epidemiology of brain metastases. Curr Oncol Rep 14:48-54, 2012” for epidemiology of brain mets in NSCLC, not reference number 4
    • This reference has been added to the introduction as well as the discussion
  • Row 61-62: therapies for metastatic NSCLC include also targeted agents.
    • Targeted atherapies have been added to the discussion Line 66-68 and discussion Line 212-223
  • Specify different incidence of brain metastases in oncogene addicted NSCLC (higher incidence in patients with EGFR, ALK, ROS1 driver).
    • There is no specific incidence report of ALK EGFR and ROS1 mutations in current literature concerning stage 1 but we included this topic in the discussion Line 212-223
  • ESMO guidelines suggest screening brain MRI with level of recommendation IIIB (Postmus PE et al. Annals of Oncology 28 (Supplement 4): iv1–iv21, 2017 doi:10.1093/annonc/mdx222).
    • We discussed this subject in the discussion Line 195-200
  • NNCN guidelines recommend brain MRI for stage II-III, while it is optional for stage IB, and not recommended for stage IA (NCCN version 3.2022)
    • We discussed this subject in the discussion Line 195-200
  • Were observed any differences among stage IA, IB and IC?
    • There were no differences between Stage IA, IB and IC. But with such a small proportion of patients with NSCLC we regrouped the patients in Stage I only.
  • Considering different incidence of brain metastases in patients with oncogene addicted NSCLC, data about oncogene drivers and PD-L1 expression are needed.
    • Unfortunately since it is a retrospective multicentric analysis not all the centers were able to provide this information, but it is a very interesting variable for a future project.
  • Reference number 6 must be updated: ESMO guidelines Annals of Oncology 29 (Supplement 4): iv192–iv237, 2018 doi:10.1093/annonc/mdy275
    • This has been modified

Reviewer 2 Report

Role of pre-operative brain Imaging in patients with NSCLC 2 Stage I: a retrospective, multicenter analysis.

Luis Filipe Azenha and colleagues provided an important retrospective, muticenter study assessing the incidence of pre-operative brain imagery (CT or RMI) in stage I NSCLC. 577 patients with baseline brain imagery were included, 8 had brain metastases.

I think that results from this large series are important. It highlights the fact that brain extension is rare in stage I NSCLC, and brain imagery should probably not be done at this disease stage.

L41 : NSCL ?

L61: add for non-oncogene addicted NSCLC. If it is not precise, authors should add targeted therapies as common treatment options.

L77: was designed ? use past

L97 : do you have an idea of the proportion of brain imaging among all-comers stage I NSCLC in your cohort ?

Table 1 : Please add the abbreviation used in the table

L109: Half of these patients had a previous malignancy in their medical history. This is an important point: do you know which kind of malignancy and when? Among patients who underwent surgical resection, does the pathology confirmed the diagnosis of lung metastasis?

How were detect brain metastases among these 8 patients: CT ou RMI?

L112-L114: I think that statistical analyses between these two population (n=8 versus v=569) have absolutely no interest. Authors should delete this part.

L148: % is missing

L164: Are you sure it is really a controversial topic? Authors should add in the text ESMO and ASCO recommendations regarding this question?

“Larger prospective studies are needed to confirm these results”. I am not sure this is truly necessary!

In the 7 centers that have included patients into this study, how is conducted the follow-up in these patients? With brain imaging ? Do you have any data regarding the incidence of brain metastases during the follow-up in stage I NSCLC ?

Author Response

Dear reviewer,

thank you for your constructive comments concerning our paper. We corrected the passages you pointed out and added the literature you suggested.

Best regards,

The authors

L41 : NSCL ?

  • Line 45 NSCLC

L61: add for non-oncogene addicted NSCLC. If it is not precise, authors should add targeted therapies as common treatment options.

  • Targeted therapies have been added to the discussion Line 66-68 and discussion Line 212-223

L77: was designed ? use past

  • Has been corrected Line 92

L97 : do you have an idea of the proportion of brain imaging among all-comers stage I NSCLC in your cohort ?

  • The centers included in this study routinely perform brain imaging in all patients with NSCLC

Table 1 : Please add the abbreviation used in the table

  • This has been added to table 1

L109: Half of these patients had a previous malignancy in their medical history. This is an important point: do you know which kind of malignancy and when? Among patients who underwent surgical resection, does the pathology confirmed the diagnosis of lung metastasis?

  • The 2 patients who received surgical resection of cerebral metastasis had pathological confirmation of lung metastasis. The other patients had a previous history of GI malignancies

How were detect brain metastases among these 8 patients: CT ou RMI?

  • 2 were diagnosed using CT scan and 6 using brain MRI

L112-L114: I think that statistical analyses between these two population (n=8 versus v=569) have absolutely no interest. Authors should delete this part.

  • We completely agree with you but we would leave the statistical analysis just to be complete

L148: % is missing

  • % was added

L164: Are you sure it is really a controversial topic? Authors should add in the text ESMO and ASCO recommendations regarding this question?

  • Since major associations including ERS don’t agree on strict guidelines whether or not brain imaging should be performed in stage I NSCLC we still consider this a controversial topic.

“Larger prospective studies are needed to confirm these results”. I am not sure this is truly necessary!

  • We believe larger studies using using driver mutations as a variable should be able to provide proper guidelines for brain imaging in early stages in order to benefit patients who suffer from cerebral metastasis at an early stage of the disease.

In the 7 centers that have included patients into this study, how is conducted the follow-up in these patients? With brain imaging ? Do you have any data regarding the incidence of brain metastases during the follow-up in stage I NSCLC ?

  • Follow-up is conducted according to ESMO guidelines

Reviewer 3 Report

Authors demonstrated the impact of brain imaging in patients with small size of NSCLC.
I would like to make some comments on this manuscript.

First of all, lung cancer with brain metastasis is not stage I. Description in the manuscript should be changed. Second, I can't imagine what kind of clinical question has been projected to be solved by this study.The frequency of brain metastasis has already been reported in the cited literature and does not contain information superior to those reports. Besides, they did not define how to confirm brain metastasis. In their cohort, 8 cases were found. Were they really brain metastasis? Brain Imaging contained CT and MRI, with or without contrast, according to their report. Were the data reliable? I'm afraid false negative can be found in cases with brain CT without contrast. 

Author Response

Dear reviewer,

thank you for your comment. We completely agree with you that NSCLC with brain metastasis is not stage I, nevertheless the whole point of this study was to determine if brain imaging is necessary for early stage lung cancer considering T and N stage (size, nodal invasion …). The current guidelines (or at least in part) dictate that NSCLC <4cm and without nodal invasion (cT2a cN0) does not require preoperative brain imaging in patients without neurological symptoms. Although previous report about the incidence of brain metastases exist in NSCLC there are only few reports about “local” stage I NSCLC and with much smaller patient cohorts  We can assure you that the data was gathered in the most reliable way and we provide information about the technical aspect in the methods section.

Best regards,

The authors.

Reviewer 4 Report

The authors try to assess the importance of pre-operative brain imaging in patients with NSCLC Stage I in a pooled cohort. The conclusion from this multicenter analysis is that brain imaging can be omitted because of the low incidence rate of 1.4%. Although the question is clinically relevant, I have major concerns about the validity of the presented data. I therefore believe that the conclusion is questionable and would ask the authors to comment on this.

Results lines 101 – 104: “469 patients (81.3%) underwent MRI, while 95 101 (16.5%) underwent CT scan. The vast majority of our cohort (569, 98.6%) had no brain metastasis.”

Since 81.3% of the patients received MRI and 16.5% received cranial CT several questions arise:

  1. Did 2.2% not have any brain imaging at all? If so, I would exclude these patients from the analysis.
  2. How were the 8 cases with brain metastases diagnosed? How many of them were diagnosed by MRI? If in these 8 patients all had MRI, then the conclusion could be that brain imaging before surgery / SABR has to be performed with this diagnostic modality.
  3. The authors state that all the patients with brain metastases had adenocarcinoma. From clinical experience it is known that adenocarcimas sometimes behave similarly to SCLC. So maybe the conclusion could be that patients with adenocarcinoma stage I should be screened with MRT because the incidence in these patients is 2% (8/400) – and maybe it would be higher if all patients with adenocarcinoma in this cohort would be screened with MRT.

Discussion 147 – 149: (…) 81.3 patients enrolled in this study underwent MRI, we consider our primary endpoint well addressed in our collective.

To me, this seems a bit far-fetched given the issues above. If almost every 5th patient either had no or insufficient (= cranial CT) imaging before surgery one cannot hold this perspective. I suggest excluding the patients that did not have any brain imaging at all or cranial CT only, which reduces the number of the cohort to 469. I also suggest that the authors delete this sentence.

Author Response

Dear reviewer,

thank you for your constructive comments concerning our paper. We corrected the passages you pointed out and added the literature you suggested.

Best regards,

The authors

The authors try to assess the importance of pre-operative brain imaging in patients with NSCLC Stage I in a pooled cohort. The conclusion from this multicenter analysis is that brain imaging can be omitted because of the low incidence rate of 1.4%. Although the question is clinically relevant, I have major concerns about the validity of the presented data. I therefore believe that the conclusion is questionable and would ask the authors to comment on this.

  • This topic was added to the discussion.

Results lines 101 – 104: “469 patients (81.3%) underwent MRI, while 95 101 (16.5%) underwent CT scan. The vast majority of our cohort (569, 98.6%) had no brain metastasis.”

Since 81.3% of the patients received MRI and 16.5% received cranial CT several questions arise:

  • Did 2.2% not have any brain imaging at all? If so, I would exclude these patients from the analysis. @Pietro
  • This was a typing mistake we corrected the numbers in the results section. We are sorry for this mistake.

  • How were the 8 cases with brain metastases diagnosed? How many of them were diagnosed by MRI? If in these 8 patients all had MRI, then the conclusion could be that brain imaging before surgery / SABR has to be performed with this diagnostic modality.
  • 2 were diagnosed using CT scan and 6 using brain MRI. Since the NPV of contrast enhanced CT scan is 94% we consider this a valid alternative technique to MRI.
  • The authors state that all the patients with brain metastases had adenocarcinoma. From clinical experience it is known that adenocarcimas sometimes behave similarly to SCLC. So maybe the conclusion could be that patients with adenocarcinoma stage I should be screened with MRT because the incidence in these patients is 2% (8/400) – and maybe it would be higher if all patients with adenocarcinoma in this cohort would be screened with MRT.

    • This is a valid point, nonetheless it has been reported that adenocarcinomas present EGFR mutations much more often than SCC for example. Recent studies also suggest that EGFR mutations increase the risk of brain metastasis. In our opinion it would be better to perform brain imaging according to EGFR mutation in the primary tumor than to histology alone. One of the pitfalls is that small tumors tend to be resected primarily, hence preoperative histology is not available. But we consider this an interesting topic for a future project.

Discussion 147 – 149: (…) 81.3 patients enrolled in this study underwent MRI, we consider our primary endpoint well addressed in our collective. 

To me, this seems a bit far-fetched given the issues above. If almost every 5th patient either had no or insufficient (= cranial CT) imaging before surgery one cannot hold this perspective. I suggest excluding the patients that did not have any brain imaging at all or cranial CT only, which reduces the number of the cohort to 469. I also suggest that the authors delete this sentence.

  • Contrast enhanced CT scan has an NPV of 94% and is in our opinion a valid alternative to brain MRI if brain MRI is not readily available.

Reviewer 5 Report

This study evaluated the incidence of brain metastasis in clinical stage I lung cancer. Although it is a multicenter study, it does not offer much new information in this regard. The authors did not have a detailed analysis of this series and the discussion was not comprehensive. Overall, the novelty is low and the context is too simplified for a high-quality journal.

  1. This series had 8 patients diagnosed with brain metastasis but only 2 patients received surgical treatment of brain metastasis. How did the authors make accurate pathologic diagnosis of the remaining 6 patients? Could the brain lesions be other etiologies rather than brain metastasis?
  2. Since the patients were recently treated, their clinical stages could be further divided into IA, IB and IC, according to tumor sizes and the AJCC 8th staging system. Did the authors further look into the data for correlation between details of clinical stages and brain metastasis?
  3. Because these patients were treated in thoracic surgery units, they should have final pathologic diagnosis. How about the pathologic stages of these patients and more important, how about pathologic stages of the 8 patients with brain metastasis?
  4. The introduction part is too short and the segmentation of the discussion part is a bit disordered.
  5. The authors used “collective” for this cohort and it is an unusual use. They should change a term.
  6. There are some typos need to be corrected. For example, in the second line of the simple summary, 230.000 is wrong.

Author Response

This study evaluated the incidence of brain metastasis in clinical stage I lung cancer. Although it is a multicenter study, it does not offer much new information in this regard. The authors did not have a detailed analysis of this series and the discussion was not comprehensive. Overall, the novelty is low and the context is too simplified for a high-quality journal.

  • Thank you for sharing your opinion. We don’t agree with this comment since to the least it confirms previous smaller studies. We are also convinced that the major variables that can be gathered by retrospective analysis are described in this paper. We can’t imagine any further analysis on a cohort of 577 patients and 8 brain metastases cases.

  1. This series had 8 patients diagnosed with brain metastasis but only 2 patients received surgical treatment of brain metastasis. How did the authors make accurate pathologic diagnosis of the remaining 6 patients? Could the brain lesions be other etiologies rather than brain metastasis?
  • Since those patients were treated by radiosurgery or other techniques we don’t have pathological confirmation. Nonetheless from a clinical point of view NSCLC metastasis is the most evident ethiology of those lesions. Brain metastasis are often located in location that are difficult to gain access to in order to perform biopsies ore even surgical resection.
  1. Since the patients were recently treated, their clinical stages could be further divided into IA, IB and IC, according to tumor sizes and the AJCC 8th staging system. Did the authors further look into the data for correlation between details of clinical stages and brain metastasis?
  • We looked into it and there was no difference. Since there were only 8 patients with brain metastasis we didn’t perform subgroup analysis.
  1. Because these patients were treated in thoracic surgery units, they should have final pathologic diagnosis. How about the pathologic stages of these patients and more important, how about pathologic stages of the 8 patients with brain metastasis?
  • All those patients were stage I as we explained under point number 2 the cohort of patients with brain metastasis is too small to be analyzed in subgroups.
  1. The introduction part is too short and the segmentation of the discussion part is a bit disordered.
  • This has been rearranged
  1. The authors used “collective” for this cohort and it is an unusual use. They should change a term.
  2. There are some typos need to be corrected. For example, in the second line of the simple summary, 230.000 is wrong.
  • Typos have been corrected. As for 230.000 we don’t understand what you mean. Please read the reference mentioned.

Round 2

Reviewer 1 Report

reference number 7 needs to be updated because there is an updated version of ESMO guidelines released in 2018 on Annals of oncology

Reviewer 4 Report

I have nothing to add.